# Differential Retinal Protein Expression in Primary and Secondary Retinal Ganglion Cell Degeneration Identified by Integrated SWATH and Target-Based Proteomics

**DOI:** 10.3390/ijms22168592

**Published:** 2021-08-10

**Authors:** Jacky M. K. Kwong, Joseph Caprioli, Ying H. Sze, Feng J. Yu, King K. Li, Chi H. To, Thomas C. Lam

**Affiliations:** 1Department of Ophthalmology, Stein Eye Institute, David Geffen School of Medicine, University of California Los Angeles, Los Angeles, CA 90095, USA; caprioli@jsei.ucla.edu; 2Centre for Myopia Research, School of Optometry, The Hong Kong Polytechnic University, Hong Kong 999077, China; 18074319r@connect.polyu.hk (Y.H.S.); yufengjuan@ykrskj.com (F.J.Y.); kk.li@polyu.edu.hk (K.K.L.); chi-ho.to@polyu.edu.hk (C.H.T.); 3Centre for Eye and Vision Science, School of Optometry, The Hong Kong Polytechnic University, Hong Kong 999077, China; 4Shenzhen Research Institute, The Hong Kong Polytechnic University, Shenzhen 518052, China

**Keywords:** neurodegeneration, ganglion cell, optic nerve, protein, proteomics, mass spectrometry

## Abstract

To investigate the retinal proteins associated with primary and secondary retinal ganglion cell (RGC) degeneration and explore their molecular pathways, SWATH label-free and target-based mass spectrometry was employed to identify the proteomes in various retinal locations in response to localized optic nerve injury. Unilateral partial optic nerve transection (pONT) was performed on adult Wistar rats and their retinas were harvested 2 weeks later. To confirm the separation of primary and secondary RGC degeneration, immunohistochemistry of RNA binding protein with multiple splicing (RBPMS) and glial fibrillary acidic protein (GFAP) was performed on retinal whole-mounts. Retinal proteomes in the temporal and nasal quadrants were evaluated with high resolution hybrid quadrupole time-of-flight mass spectrometry (QTOF-MS), and SWATH-based acquisition, and their expression was compared to the corresponding retinal quadrant in contralateral control eyes and further validated by multiple reaction monitoring mass spectrometry (MRM-MS). A total of 3641 proteins (FDR < 1%) were identified using QTOF-MS. The raw data are available via ProteomeXchange with the identifier PXD026783. Bioinformatics data analysis showed that there were 37 upregulated and 25 downregulated proteins in the temporal quadrant, whereas 20 and five proteins were upregulated and downregulated, respectively, in the nasal quadrant, respectively (*n* = 4, *p* < 0.05; fold change ≥ 1.4-fold or ≤0.7). Six proteins were regulated in both the temporal and the nasal quadrants, including CLU, GFAP, GNG5, IRF2BPL, L1CAM, and CPLX1. Linear regression analysis indicated a strong association between the data obtained by means of SWATH-MS and MRM-MS (temporal, R^2^ = 0.97; nasal, R^2^ = 0.96). Gene ontology analysis revealed statistically significant changes in the biological processes and cellular components of primary RGC degeneration. The majority of the significant changes in structural, signaling, and cell death proteins were associated with the loss of RGCs in the area of primary RGC degeneration. The combined use of SWATH-MS and MRM-MS methods detects and quantifies regional changes of retinal protein expressions after localized injury. Future investigation with this integrated approach will significantly increase the understanding of diverse processes of progressive RGC degeneration from a proteomic prospective.

## 1. Introduction

Retinal ganglion cell (RGC) degeneration is the key characteristic of optic neuropathies such as ischemic, traumatic, and hereditary neuropathies and glaucoma [1,2,3,4,5]. The progressive loss of RGCs can ultimately lead to blindness in patients. An understanding of the mechanisms that underlie the ongoing process of RGC degeneration would significantly aid in the development of new treatments to slow down the rate of disease progression or stop the degeneration altogether. However, it is challenging to study the mechanisms of RGC degeneration because the pathways leading to progressive RGC loss vary widely, both spatially and temporally; the identities of initiators, mediators, and executors that are involved in various stages of disease remain largely unknown; and their molecular interactions are complicated.

An initial injury to the central nervous system (CNS) damages the neurons, activates the glia and disrupts many physiological aspects of the tissue. These changes initiate chain reactions and mediate responses in adjacent anatomical areas [6,7,8,9,10,11]. To understand the pathways leading to the spread of degeneration, an animal model which provides a clear and predictable separation of primary and secondary degenerative events would be helpful [12,13,14]. A partial optic nerve transection (pONT) model in rats has been developed to distinguish secondary degeneration from primary RGC degeneration morphologically [15,16,17,18]. The primary lesion in the optic nerve not only displays demyelination, gliosis, and axonal loss [19] but also dramatic loss of RGC bodies at one to two weeks in the retinal area corresponding to the location of the partial optic nerve injury site [20]. After weeks or months, the gradual loss of RGC bodies occurs at locations beyond the initial injury site, which is known as secondary RGC degeneration. Although apoptosis occurs during both primary and secondary RGC degeneration through the activation of caspase 3 (Casp3), the overexpression of Bax and Bad genes, and downregulation of Bcl2 and Bcl-xl genes [21], the time course for secondary RGC body loss is considerably delayed, and the chronic disruptions to axon caliber and myelin sheath integrity are persistent for up to 6 months during secondary RGC degeneration [19]. Microarray analysis has revealed differences in innate inflammatory cell recruitment and infiltration, activation of Casp3, both intrinsic and extrinsic apoptosis-associated gene expression, and antioxidant activity between retinas with primary and secondary RGC degeneration, suggesting that their cell death involves multiple molecular events [22].

The use of a whole tissue sample for biochemical analysis masks the differential molecular signals that are specific to the primary injury site and the adjacent area. Therefore, we utilized a rat model of pONT with previously characterized RGC loss, then divided each retina into four retinal quadrants for probing the proteome profiles specific to primary and secondary RGC degeneration. Our previous work with fluorescence difference two-dimensional gel electrophoresis successfully demonstrated regional changes in protein expression in various retinal locations after localized optic nerve injury [20]. Modern liquid chromatography-mass spectrometry (LC-MS) offers a combination of exploratory protein identification with increased sensitivity, specificity, and validations for both protein quality and quantity. Bioinformatics analysis of the generated proteomic dataset provides in silico datamining for molecular networks during disease progression. For discovery, the present investigation employed an integrated approach with label-free sequential window acquisition of all theoretical mass spectra (SWATH)-MS for unbiased proteomic analysis and targeted multiple reaction monitoring (MRM)-MS for confirmation of the selected protein [23]. This experiment is an initial step, providing data for building a protein library of this rat model for future study. Overall, in this study we identified 62 proteins regulated in primary RGC degeneration and 25 proteins in secondary RGC degeneration in the retina 2 weeks after the pONT procedures, and also explored pathways associated with the processes involved in primary and secondary RGC degeneration. Among these regulated proteins, only six proteins were found to be shared with both primary and secondary RGC degeneration.

## 2. Results

The phenotypes and protein profiles of the nasal (N) and temporal (T) retinal quadrants in the experimental (left; L) eye were compared to the corresponding quadrants of the contralateral control (right; R) eye. To show RGC soma loss and glial cell reactivity after pONT, whole retinas from four animals were selected for double immunohistochemistry with antibodies against RBPMS and GFAP to confirm the location of primary and secondary RGC degeneration. Mild dropout of RBPMS-positive cells, as well as glial cells with more dendritic processes and increased GFAP-immunoreactivity, were detected in the nasal retinal quadrant after pONT (Figure 1D–F) compared to the control retina (Figure 1A–C). In contrast, a remarkable loss of RBPMS-positive cells and an extensive dendritic pattern with increased GFAP-immunoreactivity were noted in the temporal quadrant (Figure 1G–I). No noticeable difference was detected between RGC density in the temporal and nasal quadrants of control retinas (Appendix A). The density of RGCs in the temporal and nasal quadrant after pONT was lower than their corresponding quadrants in the control eyes. Similarly to our previous report [20], the current findings indicate that primary RGC degeneration occurs in the temporal quadrant, whereas secondary RGC degeneration happens in the nasal quadrant.

For proteomic analysis, each retina (*n* = 4) was divided into four retinal quadrants and only temporal and nasal retinal quadrants were elevated. The retinal quadrants collected from the experimental and control eyes were processed independently for protein extraction. Figure 2 shows the workflow for protein identification and quantification by applying untargeted and targeted approaches using Sciex Triple TOF 6600 MS and Sciex QTRAP 6500+ MS, respectively. The number of proteins and peptides identified by means of label-free SWATH-MS are summarized in Table 1. Overall, 3641 proteins (25,313 peptides) were identified for the construction of a pool spectral library. Using Peakview (version 2.2, Sciex) for MS peak extraction, about 87% of them (3173 proteins, 18,336 peptides) could be quantified across all the biological and technical replicates for the detection of significant protein changes between temporal and nasal retinal quadrants.

Volcano plots of quantified proteins expressed in the temporal and nasal retinal quadrants are shown in Figure 3A,B, respectively. The proteins identified in the temporal and nasal retinal quadrants are listed in Appendix A, respectively. According to previously published inclusion criteria [23], significant differentiation with a stringent cutoff line at a 1% false discovery rate (FDR) with *p* < 0.05 and the value of Log2 fold-change (FC) ≥ 0.43 or ≤ −0.43 was used. Comparison with the protein profiles of the experimental (left) eye and the control (right) eye revealed there were 59 and 24 regulated proteins found in the temporal and nasal retinal quadrants, respectively. Of those in the temporal quadrant, 35 were upregulated and 24 were downregulated, whereas in the nasal quadrant 19 and five proteins were upregulated and downregulated, respectively (Figure 3A,B; blue dots). Among these protein candidates, only six common proteins were found to be significantly regulated in both the temporal and nasal retinal quadrants, namely, CLU, GFAP, GNG5, IRF2BPL, L1CAM, and CPLX1 (Figure 3A,B; orange dots). L1CAM was 0.70-fold downregulated in the temporal quadrant but 1.68-fold upregulated in the nasal quadrant. The other five proteins were regulated in both retinal quadrants in a similar fashion. The details of the differentially regulated protein expression in the temporal and nasal quadrants are listed in Table 2 and Table 3, respectively.

To further validate the expression level, targeted MRM-MS was performed to evaluate 15 protein candidates of interest with a reference protein candidate, GAPDH, in each retina. The target proteins and their peptide sequences in the temporal and nasal retinal quadrants are listed in Appendix A, respectively. The transition lists for MRM-MS in the temporal and nasal retinal quadrants are shown in Appendix A, respectively. Figure 4A and Figure 5A show the fold-change of the regulated proteins in the temporal and nasal quadrants, respectively. The fold-change determined using SWATH-MS (*x*-axis) is plotted against the fold change determined using MRM-MS (*y*-axis) in Figure 4B for the temporal quadrant and Figure 5B for the nasal quadrant. Linear regression analysis for the temporal and nasal quadrants revealed a strong association between SWATH-MS data and MRM-MS data (R^2^ = 0.965 for temporal and R^2^ = 0.964 for nasal), indicating good reproducibility of the independent determination of protein fold changes in two distinct mass spectrometry systems.

To explore the potential protein–protein interaction networks, significantly regulated protein candidates were imported into the STRING database to match with genes of *Rattus norvegicus*. The “known” and “predicted” interactions for both the temporal and nasal quadrants are summarized in Figure 6. The majority of regulated proteins were shown to interact with other significantly regulated proteins, whereas there was no existing interaction evidence determined for 25 proteins. Six proteins (CLU, GFAP, GNG5, IRF2BPL, L1CAM, and CPLX1) were significantly regulated in both primary and secondary degeneration. For primary RGC degeneration, local network cluster analysis demonstrated several clusters of “co-expressed” proteins with “known interactions”, namely, intermediate filament bundle assembly and neuromodulin, annexin A2 (ANXA2) and HIRAN domain, ADF-H/gelsolin-like domain superfamily and actin, synaptic membrane and neurotransmitter release, and chemical carcinogenesis and arachidonic acid. For secondary RGC degeneration, the protein–protein interaction analysis of the regulated proteins in the nasal quadrants revealed the “co-expression” of two proteins (RGD1566320 and Pdcd6) that are related to endoplasmic reticulum–Golgi vesicular transport, and predicted that there were two groups of proteins with “known interactions” (GFAP-STAT1-L1CAM and ARAF-FGB-CLU).

A summary of bioinformatic analysis data for the temporal and nasal retinal quadrant is shown in Table 4. The PathwayGuide Analysis (Table 5; *p* < 0.05 using Fisher’s method) further predicted that top pathways in primary RGC degeneration (RT vs. LT) may be related to ferroptosis, the HIF-signaling pathway, the adipocytokine signaling pathway, necroptosis, and cell adhesion molecules, whereas secondary RGC degeneration (RN vs. LN) may be related to complement and coagulation cascades, hepatitis B, the Jak-STAT signaling pathway, pancreatic cancer, and serotonergic synapse. However, more stringent *p*-values corrected for multiple comparisons with FDR were larger than 0.05 in this analysis. Nevertheless, gene ontology (GO) analysis (Table 6) demonstrated significant changes in molecular functions, biological processes, and cellular components in primary RGC degeneration but no significant changes in secondary RGC degeneration after FDR correction of the *p*-value. These data reflect more severe molecular events at the site of primary degeneration two weeks after pONT. Significant changes involved signaling receptor binding, relating to structural constituents of the postsynaptic intermediate filament cytoskeleton in molecular functions. Furthermore, intermediate filament organization, supramolecular fiber organization, neuron projection regeneration, intermediate filament-based processes, and intermediate filament cytoskeleton organization were also shown to be involved. The cellular components further demonstrated that the involvement of the postsynaptic intermediate filament cytoskeleton, neurofibrillary tangle, axon, Schaffer collateral—CA1 synapse, and neuronal cell body are more significant in primary RGC degeneration (*p* < 0.05; *p*-value for FDR < 0.05).

## 3. Discussion

Systematic proteomic analysis can provide insights into the molecular responses of different retinal regions in response to a localized axonal injury. It is important to utilize an integrated non-targeted and targeted MS approach to study the retinal proteomes in rat retinas in an experimental model of optic neuropathy, which separates primary and secondary RGC degenerations. This study provided a comprehensive non-biased survey of the proteins involved in the primary (temporal) and secondary (nasal) loss of RGCs with a next generation data-independent SWATH-based label-free proteomics approach. Our validated proteomic data documented remarkable differences between protein expression profiles in the retinal quadrants (temporal and nasal) after pONT. The bioinformatics analysis not only revealed enriched regulated proteins that impacted significant changes in gene ontology, but also demonstrated molecular mechanisms underlying the early phase of primary and secondary RGC degeneration.

The loss of RGCs is a major characteristic of glaucoma and optic neuropathy [24,25]. Neurofilaments, tubulins, and Thy-1 membrane glycoproteins are specifically expressed by RGCs and are useful for the visualization of surviving RGCs in optic neuropathy [26,27]. Although their gene or protein regulation is under the influence of optic nerve damage [28], they are widely used as RGC markers for quantitative analysis in RGC degeneration. The present study demonstrated that the reduced levels of some structural proteins, including members of the neurofilament family (NF-L, NF-M and NF-H) and Thy-1, in the temporal retina, as well as their spatial pattern of protein expression, were consistent with the topographical quantification of RGC loss [20].

This proteomics perspective using the rat pONT model contributes considerably to the understanding of the temporal and spatial aspects of progressive RGC degeneration, because of its clear separation between primary and secondary RGC degeneration in a system-wide, unbiased investigation. Consistent with previous reports [15,16,17,18,19,20], immunohistochemical techniques with antibodies against RBPMS (RGC marker) and GFAP (glial marker) demonstrated a clear separation of extensive neuronal loss in the region with primary RGC degeneration (temporal) and mild loss in the area of secondary RGC degeneration (nasal) after pONT. In contrast, increased glial reactivity was uniform over the entire retina. The same pattern of GFAP and neurofilament (NEFL, NEFM, and NEFH) regulation was noted in the retinal quadrants, as evaluated by SWATH-MS techniques. Neurofilaments (NF) are intermediate filaments found in the cytoplasm of neurons and form the neuronal cytoskeleton, along with microtubules and microfilaments. Degradation and loss of NFs indicates the loss of RGCs. Dephosphorylation of the heavy neurofilament subunit was recently reported to be a key degenerative process in the monkey chronic glaucoma model [29], but had not been detected in a murine model at this early time point. GFAP is an intermediate filament protein, expressed by astrocytes in the CNS [30], which is usually responsible for cell communication and the functioning of the blood–brain barrier. In general, GFAP is recognized as a biomarker for many internal and external stimuli, as well as neurodegenerative diseases in the CNS. Similarly, increased levels of GFAP were found in the retinas of primate and rodent eyes with optic nerve transection and IOP elevation, as well as in patients with glaucoma [31,32]. A microarray analysis demonstrated that the gene expression of GFAP in the secondary degeneration was decreased at day one and remained unchanged 7 days after pONT, but qRT-PCR analysis showed no change in the GFAP level and an increase at 7 days, indicating a discrepancy due to the filtration process in the pair-wise comparison [22]. In our study, we observed a consensus between the data obtained from both immunohistochemical and MS techniques, supporting the use of this experimental design to investigate the molecular processes for primary and secondary RGC degeneration.

In addition to significant changes in filament organization, our study also demonstrated that alterations in glutathione metabolism and mitochondrial dysfunction-associated proteins, activation of the coagulation cascade, and upregulation of cholesterol transport proteins exhibited similarities with the rat model of glaucoma analyzed using a multiplexed quantitative proteomics approach [33]. Numerous regulated proteins shared by the two rat models of pONT and IOP elevation via microbead injection include AHNAK, APOE, FGB, KNG1, NDUFA2, GSTM5, and members of S100, crystallin, CPLX, and ANXA families, providing evidence that the mechanisms underlying the progressive RGC degeneration after pONT injury are relevant to glaucoma.

The MS technology with label free SWATH acquisition detected 3641 proteins in the adult Wistar rat retina. Of these, 3174 proteins were selected for quantification based on the experimental criteria (six ion transitions per peptide, 90% peptide confidence threshold, 1% FDR; data shown in Appendix A). A protein library database was established to explore the pathways specific to the early stages of primary and secondary RGC degeneration by the present study and will also be used for future experiments of protein profiling in the later stages of degeneration. The SWATH-MS is the only approach capable of multiplexed protein quantification from a proteome perspective, with added features of discovery, unbiased target selection, antibody-free identification and quantification, and inter-center reproducibility [34], making it an indispensable tool for high-throughput proteomics experiments.

Independently, targeted MRM-MS was used to evaluate ten regulated proteins in the temporal quadrant and five proteins in the nasal quadrants, with reference proteins in both quadrants. The goal was to validate the differential protein expressions detected with SWATH-MS in various retinal regions by means of a robust quantitative method. The linear regression analysis for the proteins expressed in both the temporal and nasal quadrants demonstrated a strong agreement between these two techniques. Therefore, we believe that the combination of both label-free and targeted MS techniques offers a valid, high-throughput workflow for protein mining and independent experiment setups, and eliminates the technical issues raised by other methods such as immunohistochemistry, Western blotting, and polymerase chain reaction (PCR). The majority of these latter methods are only semi-quantitative and their applications are largely influenced by their limited sensitivity and the specificity of reagents, antibodies, and the sample volume [35].

Proteomic analysis was used to explore the differential expressions of 59 and 24 proteins in the temporal and nasal retinal quadrants, respectively, 2 weeks after injury. Six of these proteins, CLU, GFAP, GNG5, IRF2BPL, L1CAM and CPLX1, were regulated in both regions. How different RGC compartments deteriorate in primary and secondary RGC degeneration is not completely understood [36]. Interestingly, the findings show that CPLX I, one of the SNARE-regulating proteins, which regulates a late step in the release process and which maintains synaptic vesicles for the transferring of visual information, is downregulated in both the temporal and nasal retina to similar extents (FC 0.62 vs. 0.66 when compared to controls). In the mammalian retina, CPLX I is expressed by the somata of displaced amacrine cells and RGCs in the RGC layer, as well as the somata of amacrine cells in the inner nuclear layer [37]. Whether amacrine cells are involved in both primary and secondary RGC degeneration at the same time or whether the loss of synaptic vesicles precedes the degradation of structural proteins needs to be clarified. In contrast, the levels of L1CAM, a neuronal cell adhesion molecule in the temporal and nasal retina, are significantly altered in opposite directions. It is unclear whether the elevated level of L1CAM has an implication in neurite growth or cell adhesion before RGC bodies are lost [38,39]. Further investigation to delineate the sequence of compartmental degeneration is necessary in order to develop new neuroprotective treatments by targeting different cellular components.

It is important to note that most of the shared proteins (regulated in both the temporal and nasal retinal quadrants) are associated with development and cell death. Clusterin (CLU) is a multifunctional, disulfide-linked glycoprotein and its mRNA is present in almost all mammalian tissues. It is expressed constitutively and developmentally in epithelial cells for promoting aggregation and cell adhesion. In the retina, clusterin is shown to play a role in the protection of the blood–retina barrier and photoreceptors during age-related macular degeneration (ARMD), diabetic retinopathy, ischemic retinopathy, and other diseases [40,41]. IRF2 binding protein 2 (IRF2BP2) plays a role in the development of the CNS and in neuronal maintenance as a transcriptional regulator. Cruz et al. (2017) showed that IRF2BP2 expression in macrophages/microglia reduced inflammatory cytokine expression after photothrombotic stroke surgery [42]. GNG5 controls neuronal migration and development of the cortex through various G-protein-coupled receptors. The overexpression of Gng5 has been related to apoptosis in adenocarcinoma and invasive ductal carcinoma and is considered to be a prognostic indicator of gliomas [43].

The present study provides evidence that the generation of reactive oxygen species (ROS) due to lipid peroxidation may be a key element in the mechanism of primary RGC degeneration. Firstly, analysis of the data in the context of pathways obtained from the Kyoto Encyclopedia of Genes and Genomes (KEGG) database showed that ferroptosis may be involved in the pathway leading to primary RGC degeneration. Ferroptosis is an iron-dependent non-apoptotic programmed necrosis, which is initiated by iron accumulation and overload, causing lipid ROS and cellular membrane damage. This novel programmed cell death has been recently implicated in retinal degeneration [44,45]. Secondly, there was a cluster of protein–protein interactions of regulated proteins, including serotransferrin (TF), ceruloplasmin (CP), alpha-1-macroglobulin (A1M/PZP), alpha-2-macroglobulin (P06238), transthyretin (TTR), high-molecular-weight kininogen (KNG1), and apolipoprotein E (APOE), suggesting the possible imbalance of ROS due to vascular and inflammatory alteration in the retina [46]. Thirdly, the upregulation of aldehyde dehydrogenase 3 family member A1 (ALDH3A1) suggests an endogenous defensive response to a toxic environment. ALDH3A1 can oxidize various aldehydes to their corresponding acids and is involved in the detoxification of alcohol-derived acetaldehyde and metabolism of corticosteroids, biogenic amines, neurotransmitters, and lipid peroxidation. The expression of ALDH3A1 is suggested to play an important role in the stress response and the anti-oxidation pathway for the protection of RGCs, transforming growth factor-beta in the treatment of diabetic retinopathy [47,48] and corneal cells against ultra-violet damage [49]. Our local network clusters analysis also indicated that there is an interaction between Aldh3a1, Oplah, and Mgst3, which are also believed to be related to the glutathione metabolic process.

The observation of secondary RGC degeneration revealed three groups of protein–protein interactions that were not shared with primary RGC degeneration. First, the co-expression of RGD1566320 (small integral membrane protein 26, SMIM26) and Pdcd6 (programmed cell death 6) were decreased, indicating an alteration in the endocytosis of cell surface receptor trafficking and cell death machinery. PDCD6, a calcium binding protein, is required in order to form a complex with PDCD6 interacting protein (PDCD6IP) as a regulator of cell death, controlling both caspase-dependent and -independent pathways [50]. This complex was shown to interact with pro-caspase-8 and TNF alpha receptor 1 (TNF-R1), which is involved in the death of motor neurons and represents a critical link between the endo-lysosomes and signaling or an execution step in neuronal death [51]. Second, there is a predicted interaction between the upregulation of GFAP, STAT1, and L1CAM. Yang et al. demonstrated that *stat1* gene, the transcription factor that plays a role in promoting apoptotic cell death by mediating proapoptotic activities of cytokines, was only significantly increased one day after IOP elevation in an experimental glaucoma model but not in the optic nerve transection model [52]. Agudo et al. showed that the STAT1 protein was expressed in the inner retinal layers of the control retina and its level increased after optic nerve transection and crush from 48 h up to 7 days [53]. It would be interesting to investigate whether sustained STAT1 signaling is required for the cell adhesion and glial activation that leads to the spread of progressive RGC death. Lastly, there was increased expression of ARAF, fibrinogen beta chain (FGB) and CLU. RAF kinases are major constituents of the mitogen-activated signaling pathway, regulating cell proliferation, differentiation, and cell survival [54,55]. However, the function of ARAF in the retina remains unclear. Differential patterns of distribution and subcellular localization of ARAF in the retina will help to understand its distinct signaling regulatory function. Based on the evidence of the upregulation of FGB and CLU, it is tempting to hypothesize that neuroinflammation and vascular dysfunction is a key event in secondary RGC degeneration.

In summary, we performed non-biased SWATH proteomic analysis with validation with targeted MRM-MS of the retina two weeks after pONT and established a protein library database of this experimental model for future investigations. Correlated with a dramatic loss of RGCs in the area of primary RGC degeneration and glial activation over the entire retina, the current findings provide an integrated view of differential molecular changes in the early stage of primary and secondary RGC degeneration. There were more proteins regulated in the temporal retinal quadrant than in the nasal quadrant. Strong association between the data obtained with SWATH-MS and MRM-MS supports the idea that this combination of identification and validation is a powerful tool for the exploration of spatial protein expression during progressive RGC degeneration. Significant changes of structural proteins for intermediate filament assembly and organization and sub-cellular component alterations, including those of the cell body, axon, dendrite, and synapse, in primary RGC degeneration are related to ROS overload and lipid peroxidation. Future research efforts using this integrated proteomic approach are warranted to investigate the diverse processes of progressive RGC degeneration.

## 4. Materials and Methods

### 4.1. Animals and Partial Optic Nerve Transection (pONT)

All animal procedures were performed in accordance with the ARVO statement for the Use of Animals in Ophthalmic and Vision Research and policies of the UCLA Animal Research Committee. Adult Wistar rats (3 months-old; 300–350 g) were housed with standard food and water provided ad libitum in the animal research facility of the University of California Los Angeles and kept for at least one week before surgical procedures. After anesthesia with isoflurane gas and topical 1% proparacaine eye drops, an incision was made in the temporal conjunctiva for access to the retrobulbar optic nerve, using a diamond knife to incise the optic nerve to a depth of one third of its diameter 2–3 mm behind the globe, following our previously published procedure [20]. Following conjunctival suturing, ophthalmoscopic examination was performed to ensure complete retinal blood flow. Prophylactically, topical tobramycin was applied immediately and then twice daily for 2 days. Animals were sacrificed via a carbon dioxide overdose two weeks after pONT with cervical dislocation as the secondary euthanasia method to ensure complete euthanasia. For each animal, surgical procedures were performed on one eye, whereas the contralateral eye served as an untreated control. Both eyeballs were enucleated for the following analyses. Aiming to define the difference between primary and secondary RGC degeneration after pONT, the experiments were carried out to compare the morphological and molecular changes only in the temporal and nasal retinal quadrants.

### 4.2. Double Immunohistochemistry on Retinal Whole-Mounts

To verify the location of morphological changes resulting from pONT, animals were randomly selected for this experiment. Enucleated eyeballs were fixed with 4% paraformaldehyde in 0.1 M phosphate buffer for 1 h and the whole retinas were dissected and processed as described previously [27]. Briefly, the samples were incubated with 10% fetal bovine serum for 1 h to block non-specific staining and then in a solution of RBPMS (1:500; rabbit; ProSci, Polway, CA, USA) and GFAP (1:1000; chicken; Abcam, Waltham, MA, USA) antibodies in PBS containing 1% Triton, 0.5% bovine serum albumin (BSA), and 0.9% sodium chloride (PBS-T-BSA) overnight at 4 °C. After washing in fresh PBS-T-BSA, the retinas were incubated with secondary Alexa Fluor 488 goat anti-rabbit IgG antibody (1:1000) and Alexa Fluor 647 goat anti-chicken IgG antibody (1:1000) overnight at 4 °C. Following radial cuts, the retinas were mounted on a glass slide and all four retinal quadrants (superior, temporal, inferior, and nasal) were clearly labeled for microscopic examination. The pattern of loss of RBPMS-positive cells and GFAP immunoreactivity was evaluated in a masked fashion [27].

### 4.3. Retinal Sample Homogenization and Protein Extraction

Four rat retina tissues (rat number 203,204, 205, and 206) were used for this study. The dissected retina tissues were separated into four regions (Superior, Inferior, Nasal, Temporal). Only the temporal and nasal retinal quadrants were homogenized with Precellys 24 homogenizer (Bertin Technologies, Aix-en-Provence, France) with 100 µL lysis buffer containing 7 M urea, 2 M thiourea, 30 mM Tris, 0.2% (*w*/*v*) Biolytes, 1% (*w*/*v*) dithiothreitol, 2% (*w*/*v*) CHAPS, and 1% (*w*/*v*) ASB14 in protease inhibitor cocktail (Roche Applied Science, Basel, Switzerland) at 5800 rpm twice for 30 s with a 20-s interval. The homogenized protein samples were centrifuged at 218,000× *g* for 25 min at 4 °C. The supernatant was collected for protein concentration measurements using the Bradford assay (Bio-Rad, Hercules, CA, USA).

### 4.4. Protein Reduction, Alkylation, and Digestion

Based on the protein concentration, 70 µg of protein from each sample was used for further analysis. The samples were reduced in 10 mM dithiothreitol (DTT) in 25 mM NH_4_HCO_3_ for 1 h at 37 °C, followed by alkylation in 20 mM iodoacetamide (IAA) in 25 mM NH_4_HCO_3_ for 30 min at room temperature in the dark. The protein was then precipitated by adding four volumes of ice-cold acetone and stored at −20 °C overnight. The samples were centrifuged at 218,000× *g* for 25 min at 4 °C. After supernatant removal, the protein pellet was washed with 80% ice-cold acetone and the supernatant was removed again. The pellet was re-dissolved in 8 M urea and gradually diluted to 1 M urea with 25 mM NH_4_HCO_3_. Trypsin with a final ratio of 1:25 (trysin: protein amount, *w*/*w*), was added and the mixture incubated at 37 °C overnight for in-solution digestion.

Peptides were purified before injection to a mass spectrometer using an Oasis HLB cartridge (Waters Associates, Inc., Milford, MA, USA). In brief, the cartridge was conditioned with 1 mL acetonitrile (ACN), followed by 1 mL 0.1% trifluoracetic acid (TFA) for equilibration. Each peptide sample was added with 900 µL 0.1% TFA) and bound to the cartridge by passing it slowly (1 drop/second) into the cartridge (sample load) two times. The cartridge was washed using 1 mL 0.1% TFA and consequently 0.1% TFA in 5.0% methanol. The bound peptides were eluted by slowly passing 0.5 mL 0.1%TFA with 60% ACN through the cartridge medium, which was dried by means of a vacuum centrifuge. Finally, the purified peptides were re-constituted in 20 µL 0.1% formic acid (FA). The peptide concentration after this purification in each sample was measured again using the Pierce Quantitative Colorimetric Peptide Assay (catalog #23275, Thermo Scientific, Rockford, IL, USA). The peptide concentration was adjusted to 0.5 µg/µL. A pool of sample (3 µg) from individual biological samples was used for protein identification as a reference library. A second sample (2 µg) was quantified, using SWATH acquisition in duplicate. The mass spectrometry proteomics data have been deposited into the ProteomeXchange Consortium via the PRIDE [56] partner repository with the dataset identifier PXD026783.

### 4.5. Sciex TripleTOF 6600 Mass Spectrometry and SWATH-MS Acquisition

Untargeted protein identification and SWATH-MS data were acquired on a TripleTOF 6600 system (Sciex, Darmstadt, Germany) with Analyst TF (version 1.7) fitted with a Nanospray III source and equipped with a hybrid quadrupole time-of-flight mass analyzer. A peptide sample was loaded onto a trap column (350 µm × 0.5 mm, C18) by loading the buffer (0.1% FA, 5% acetonitrile) at a speed of 2 µL/min for 15 min. It was then separated on a nano-LC analytical column (100 µm × 30 cm, C18, 5 µm) using an Ekisgent 415 nano-LC system. MS data were searched for on the UniProt database by ProteinPilot^TM^ (version 5.0, Sciex) software using the Paragon algorithm. For ProteinPilot, queries were matched to rat (*Rattus norvegicus*), tryptic digested peptides, and alkylation (iodoacetamide) and only proteins with 1% global false discovery rate (FDR) were considered as true positive identifications.

For protein quantification, all SWATH-MS acquisitions were conducted in duplicate for each biological sample. A variable isolation window (*n* = 100) in a mass range of *m*/*z* 100–1800 was used. An information-dependent acquisition (IDA) was imported as the reference ion library. Both IDA and SWATH data were processed with PeakView (version 2.2, Sciex). Proteins quantified with up to 30 peptides per protein, 6 transitions per peptide, and a 90% peptide confidence threshold at 1% FDR within a 10 min XIC peak elution window were selected for further MarkView (version 1.3, Sciex) data processing. Samples were normalized by most likely ratio (MLR) normalization with a paired *t*-test used to detect significantly differentiated proteins in biological samples among experimental groups. Statistically significant differential proteins detected using SWATH-MS of two groups were further filtered and validated using MRM-MS.

### 4.6. MRM-MS Assay of Differentially Expressed Proteins

The BLAST information of differentially expressed proteins was obtained to establish MRM methods. All MRM-MS data were acquired using a QTRAP 6500+ mass spectrometer (Sciex, Concord, ON, Canada) fitted with an electrospray ionization source (Sciex, Concord, ON, Canada) and operated in positive-ion scan mode. Typical parameters were set as follows: curtain gas (CUR), 25; collision gas (CAD), 9; spray voltage (IS), 5500 V; temperature (TEM), 500 °C; ion-source gas 1 (GS1), 50; ion-source gas 2 (GS2), 60. The entrance potential (EP) was set to 10, and the collision cell exit potential (CXP) was set to 7. The declustering potential (DP) was calculated via linear regression: DP = 0.0729 × (*m*/*z*) + 31.117. The collision energy (CE) linear regression for doubly charged peptide: CE = 0.036 × (*m*/*z*) + 8.857; triply charged peptide: CE = 0.0544 × (*m*/*z*) − 2.4099. For each analysis, 3 ug of homogenized, tryptic-digested retinal tissue was loaded onto a Waters ACQUITY UPLC CSH C18, 1.7 um, 2.1 × 100 mm column fitted with a Waters Acquity UPLC CSH C18, 1.7 um, 2.1 × 5 mm pre-column using an Eksigent ultraLC 110 system (Sciex, Concord, ON, Canada). The sample was separated at a flow rate of 200 uL/min, using the following gradient: 0–2 min maintaining 5% solvent B (98% acetonitrile/0.1% formic acid), 2.0–32.0 min ramping solvent B 5%–35%, 32.0–35.0 min ramping solvent B 35%–50%, 35.0–35.5 min ramping solvent B 50%–90%, 35.5–40.0 min maintaining solvent B at 90%, 40–40.5 min decreasing solvent B 90%–5% and maintaining for 7.5 min. Solvent A was 95% water/0.1% formic acid.

Target proteins were filtered with the following criteria: amino acid length between 8–25 units, unique peptide sequence, excluding peptides with structural modifications and missed tryptic cleavage. Proteins with less than three signature peptide sequences were not considered. The high-quality peptide transitions were further refined with retention linear regression, r > 0.95, and a similarity score, dotp > 0.9. The MRM data were analyzed with Skyline and MultiQuant^TM^ (version 3.03, Sciex). The raw data were searched against the corresponding protein FASTA sequence and matched with a previously acquired TOF-MS spectral library. The MRM transition quality was determined using the dotp score of the peak area and retention time. The MRM raw data were further processed using the MultiQuant^TM^ software and the MQ4 algorithm for automatic peak integration and the resulting retention time, peak area, peak height and signal-to-noise ratio for each monitored ion transition. The weighted average analysis was performed as previously reported [57]. In brief, peaks were normalized by calculating the peak area for each transition relative to the peak area of GAPDH transition. The transition response was calculated from the average of the three technical replicates of individual biological samples and type (treatment, control). The transition weighted average was calculated respective to the weight = 0 for S/N ratio < 5 and weight = 1 for S/N ratio ≥ 20. A Sigmoid distribution was applied to the intermediate S/N value. The weighted fold change for peptides was calculated as the weighted average of its transitions. The weighted fold change for proteins was further calculated as a weighted average of the corresponding individual peptides. The reported standard deviation was calculated based on the weighted average protein fold changes of individual biological samples. A Welch’s *t*-test (unequal variances *t*-test) *p*-value < 0.05 was considered statistically significant in all cases. The experimental design is shown in Figure 2.

### 4.7. Pathway Analysis

To explore the pathways involved in primary and secondary RGC degeneration, significantly differentially expressed proteins in temporal and nasal retinal quadrants of the experimental eye were further analyzed using a cloud-based iPathwayGuide platform (https://www.advaitabio.com/ipathwayguide; 2021-Mar-25) which adopted a proprietary impact analysis approach that was found to be useful in retinal proteomics studies [58,59]. A threshold of 0.05 for statistical significance (*p*-value) and a log fold change of expression with an absolute value of at least 0.43 was employed. The data were analyzed in the context of pathways obtained from the Kyoto Encyclopedia of Genes and Genomes (KEGG) database (Release 96.0+/11-21) [60,61] and gene ontologies from the Gene Ontology Consortium database (14 October 2020) [62,63].

## 5. Conclusions

Taken together, the presented proteomic profiling and analysis data confirm the differential molecular responses between the neural tissue that is directly injured and the neighboring tissue that is vulnerable to secondary degeneration. The data provide new clues about the molecular events and cellular components responsible for primary and secondary degeneration. Findings of this study encourage further research to better understand the key players involved in different stages of progressive RGC degeneration which is initiated by ischemia, trauma, and IOP-related injury, and to develop new therapies for neural repair and neuroprotection.

## Figures and Tables

**Figure 1 ijms-22-08592-f001:**
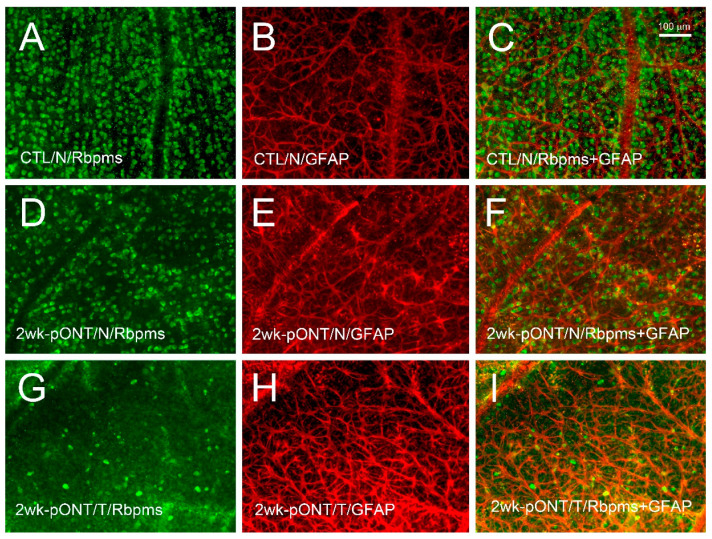
Loss of RGCs and glial reactivity two weeks after pONT. Double immunohistochemistry of RBPMS and GFAP was performed on retinal wholemounts. Compared to the contralateral control (**A**–**C**), noticeable dropout of RGCs was detected and reactive astrocytes were present in the nasal quadrant after pONT (**D**–**F**). A more dramatic loss of RGCs and increased astrocytic reactivity were noted in the temporal quadrant after pONT (**G**–**I**). N = nasal; T = temporal; RBPMS = RNA binding protein with multiple splicing; GFAP = glial fibrillary acidic protein; 2 wk = 2 weeks; pONT = partial optic nerve transection; CTL = control.

**Figure 2 ijms-22-08592-f002:**
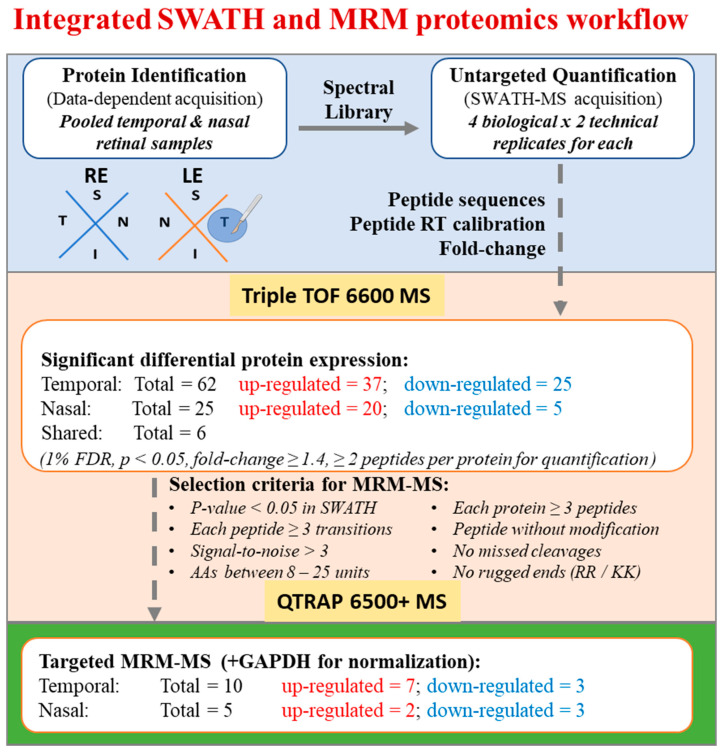
Workflow for integrated SWATH and MRM proteomics.

**Figure 3 ijms-22-08592-f003:**
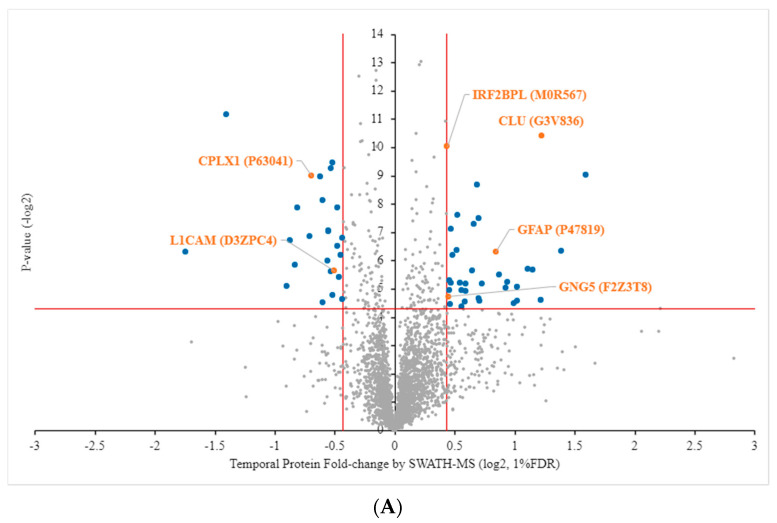
Volcano plot of proteins quantified using SWATH-MS. A plot of protein fold-change (log2, *x*-axis) and *p*-values (-log2, *y*-axis). Significantly differentiated unique proteins in quadrant positions (blue) and shared proteins (orange) were determined by *p*-values < 0.05, fold-change log2 > 0.43 (up-regulation) or ≤ −0.43 (down-regulation) and a 1% false discovery rate (FDR). (**A**) Left temporal (LT) compared to right temporal (RT). (**B**) Left nasal (LN) compared to right nasal (RN). Names of shared proteins are shown.

**Figure 4 ijms-22-08592-f004:**
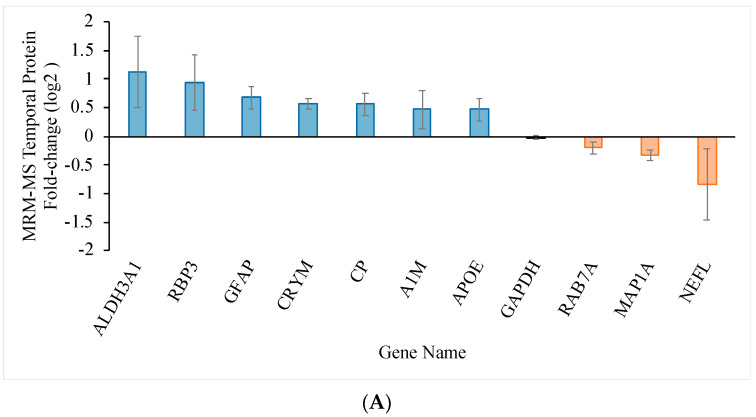
Relationship between protein quantification using SWATH-MS and targeted MRM-MS approaches in the temporal retinal quadrant two weeks after pONT. (**A**) Significantly differentiated retinal proteome in the temporal quadrant after pONT compared to control. The bar chart shows the average protein fold change with standard deviation (*n* = 4, *p* < 0.05, 1% FDR). (**B**) The retinal proteome fold-change correlation in the temporal quadrant 2 weeks after pONT treatment, measured using independent MRM-MS and SWATH-MS experiments with high regression (R^2^ = 0.965) and Pearson’s correlation (R^2^ = 0.958) on differentially expressed proteome. GAPDH was included as a reference. (**C**) Retention time linear regression plot of total 99 MRM transitions, governing 33 peptides used for the identification and fold-change calculation of 11 temporal proteins (*n* = 4, in triplicate).

**Figure 5 ijms-22-08592-f005:**
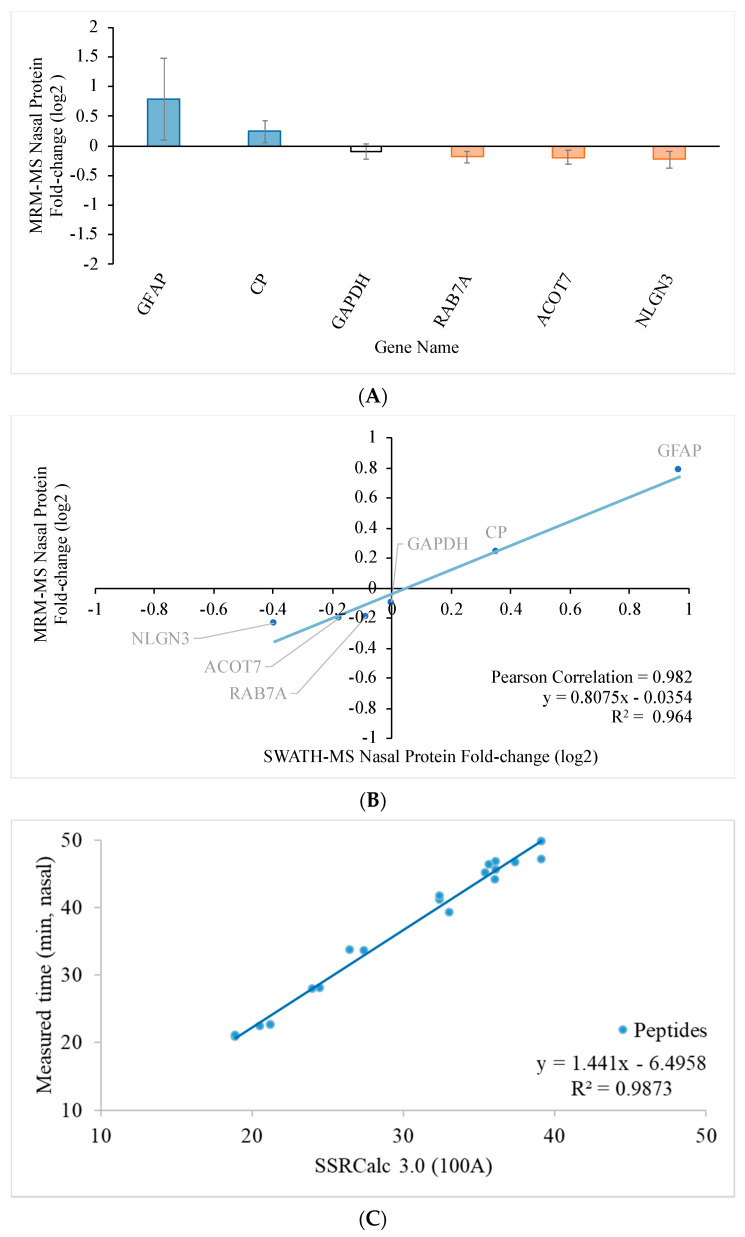
Relationship between protein quantification using targeted SWATH-MS and MRM-MS approaches in the nasal retinal quadrant two weeks after pONT. (**A**) Significantly differentiated retinal proteome in the nasal quadrant after pONT compared to control. The bar chart shows the protein fold change with standard deviation (*n* = 4, *p* < 0.05, 1% FDR). (**B**) The retinal proteome fold-change correlation in the nasal quadrant 2 weeks after pONT treatment measured by independent MRM-MS and SWATH-MS experiments with high regression (R^2^ = 0.964) and Pearson correlation (R^2^ = 0.982) on differentially expressed proteome. GAPDH was induced as a reference. (**C**) Retention time linear regression plot of total 51 MRM transitions, governing 17 peptides used for the identification and fold-change calculation of six nasal proteins (*n* = 4, in triplicate).

**Figure 6 ijms-22-08592-f006:**
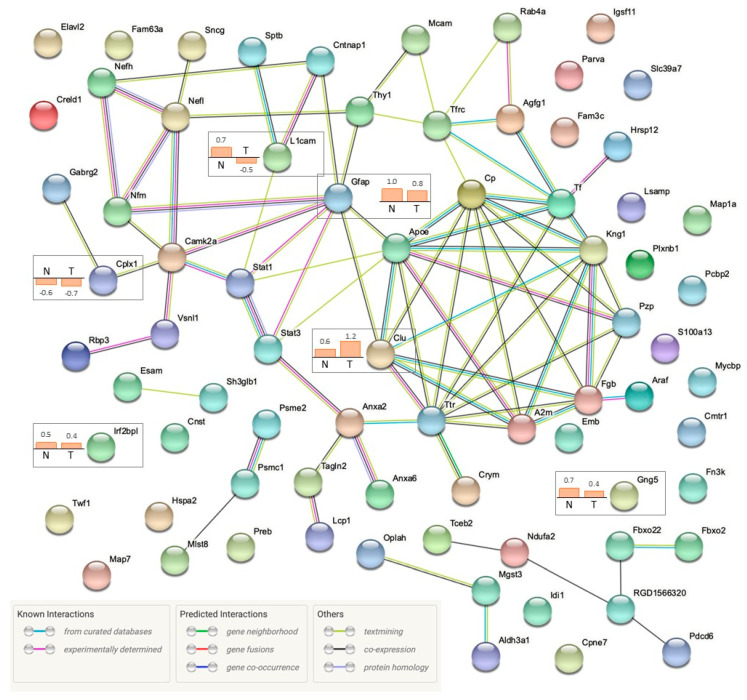
Protein–protein interaction analysis of regulated proteins two weeks after pONT with STRING database (STRING, version 11.0). A total of 80 non-redundant, significantly differentiated proteins from the nasal or temporal quadrants were converted to 75 matched gene IDs from UniProt accession numbers using the Uniprot database. Protein–protein interaction network was executed with “*Rattus norvegicus*” as the selected organism. Shared proteins (*n* = 6) are highlighted with their fold change (log2) determined by SWATH-MS.

**Table 1 ijms-22-08592-t001:** Protein and peptide identification in retinal samples 2 weeks after pONT.

	Sample Number				Pooled Sample *	MarkerView **
	1	2	3	4		
Protein	3560	3270	3504	3587	3641	3173
Peptide	24,310	20,800	23,569	25,080	25,313	18,336

* Number of proteins and peptides identified in the combined spectral library at the 1% global FDR level. ** Number of proteins and peptides extracted using MarkerView software for protein quantitation.

**Table 2 ijms-22-08592-t002:** Protein regulation in the temporal quadrant 2 weeks after pONT compared to contralateral control (*p* < 0.05 and fold-change log2 > 0.43 (upregulation) or ≤ −0.43 (downregulation)). R = right; L = left; FC = fold change.

Uniprot	Protein Name	FC L/R	LogFC	*p*-Value
Upregulation				
M0R9G2 (P06238)	Alpha-2-macroglobulin	3.02	1.59	0.00
P11883	Aldehyde dehydrogenase, dimeric NADP-preferring (EC 1.2.1.5) (Aldehyde dehydrogenase family 3 member A1) (HTC-ALDH) (Tumor-associated aldehyde dehydrogenase)	2.61	1.39	0.01
G3V836	Clusterin	2.33	1.22	0.00
P01048	T-kininogen 1 (Alpha-1-MAP) (Major acute phase protein) (T-kininogen I) (Thiostatin) (cleaved into: T-kininogen 1 heavy chain (T-kininogen I heavy chain); T-kinin; T-kininogen 1 light chain (T-kininogen I light chain))	2.33	1.21	0.04
P52631	Signal transducer and activator of transcription 3	2.22	1.15	0.02
A0A0G2QC06	Serotransferrin	2.16	1.11	0.02
F1LM19	Alpha-2-HS-glycoprotein	2.03	1.02	0.03
D3ZVR7	Prostamide/prostaglandin F synthase (Prostamide/PG F synthase) (Prostamide/PGF synthase) (EC 1.11.1.20)	2.03	1.02	0.04
D3ZS58	NADH dehydrogenase (ubiquinone) 1 alpha subcomplex subunit 2	1.98	0.99	0.04
P02767	Transthyretin (Prealbumin) (TBPA)	1.92	0.94	0.03
G3V7F1	G protein beta subunit-like, isoform CRA_a (Target of rapamycin complex subunit LST8)	1.90	0.92	0.03
G3V8Q4	Retinol binding protein 3, interstitial (Retinol-binding protein 3)	1.83	0.87	0.02
P47819	Glial fibrillary acidic protein (GFAP)	1.79	0.84	0.01
A0A0G2JUT0	Heat shock-related 70 kDa protein 2	1.65	0.73	0.03
A0A0G2K151	Apolipoprotein E	1.63	0.70	0.04
Q63041	Alpha-1-macroglobulin (Alpha-1-M) (Alpha-1-macroglobulin 165 kDa subunit) (cleaved into: Alpha-1-macroglobulin 45 kDa subunit)	1.62	0.70	0.04
Q9QYU4	Ketimine reductase mu-crystallin (EC 1.5.1.25) (CDK108) (NADP-regulated thyroid-hormone-binding protein)	1.62	0.70	0.01
Q07936	Annexin A2 (Annexin II) (Annexin-2) (Calpactin I heavy chain) (Calpactin-1 heavy chain) (Chromobindin-8) (Lipocortin II) (Placental anticoagulant protein IV) (PAP-IV) (Protein I) (p36)	1.61	0.68	0.00
Q5XFX0	Transgelin-2	1.58	0.66	0.01
G3V7K3	Ceruloplasmin (Ceruloplasmin, isoform CRA_a)	1.56	0.64	0.02
Q4V8F6	Pcbp2 protein (Poly(rC)-binding protein 2)	1.51	0.59	0.03
Q5RJR2	Twinfilin-1	1.50	0.59	0.03
A0A0G2KB52	Microtubule-associated protein 7	1.50	0.56	0.04
M0R9D5	“Obsolete”	1.47	0.55	0.03
D3ZTB5	S100 calcium-binding protein A13	1.47	0.55	0.05
G3V818	Alpha-parvin (Parvin, alpha, isoform CRA_a)	1.45	0.54	0.03
Q5XI38	Lymphocyte cytosolic protein 1	1.44	0.53	0.01
H1UBN0	Copine-7 (Copine VII)	1.43	0.52	0.01
P62870	Elongin-B (EloB) (Elongin 18 kDa subunit) (RNA polymerase II transcription factor SIII subunit B) (SIII p18) (Transcription elongation factor B polypeptide 2)	1.39	0.48	0.01
Q63798	Proteasome activator complex subunit 2 (11S regulator complex subunit beta) (REG-beta) (Activator of multicatalytic protease subunit 2) (Proteasome activator 28 subunit beta) (PA28b) (PA28beta)	1.38	0.47	0.01
P97608	5-oxoprolinase (EC 3.5.2.9) (5-oxo-L-prolinase) (5-OPase) (Pyroglutamase)	1.38	0.47	0.03
F1M9N7	Arf-GAP domain and FG repeat-containing protein 1	1.37	0.46	0.05
D4A7F2	Myc-binding protein (RCG31143)	1.37	0.45	0.03
P52759	2-iminobutanoate/2-iminopropanoate deaminase (EC 3.5.99.10) (Liver perchloric acid-soluble protein) (L-PSP) (Reactive intermediate imine deaminase A homolog) (Translation inhibitor L-PSP ribonuclease) (EC 3.1.-.-) (UK114 antigen homolog) (rp14.5)	1.37	0.45	0.03
F2Z3T8	Guanine nucleotide-binding protein subunit gamma	1.36	0.45	0.04
M0R567	Interferon regulatory factor 2-binding protein-like	1.35	0.44	0.00
A1L1L5	Ccnk protein (Cyclin K)	1.35	0.43	0.00
Downregulation				
D4ADS8 (Rab4a)	Ras-related protein Rab-4A	0.74	−0.43	0.01
G3V6W2	Prolactin regulatory element-binding protein (RCG62389, isoform CRA_a)	0.74	−0.44	0.04
D4ADS4	Microsomal glutathione S-transferase 3 (RCG46430)	0.72	−0.45	0.01
P11275	Calcium/calmodulin-dependent protein kinase type II subunit alpha (CaM kinase II subunit alpha) (CaMK-II subunit alpha) (EC 2.7.11.17)	0.72	−0.46	0.02
Q6XDA0	Spectrin beta chain	0.72	−0.48	0.00
Q6IMZ3	Annexin	0.70	−0.48	0.01
D3ZPC4	Neural cell adhesion molecule L1	0.70	−0.51	0.02
Q32Q88	F-box protein 22 (RCG58340, isoform CRA_b)	0.70	−0.52	0.00
Q4V7F2	Cysteine-rich with EGF-like domain protein 1	0.70	−0.52	0.04
O09032	ELAV-like protein 4 (Hu-antigen D) (HuD) (Paraneoplastic encephalomyelitis antigen HuD)	0.69	−0.53	0.00
P97846	Contactin-associated protein 1 (Caspr) (Caspr1) (Neurexin IV) (Neurexin-4) (Paranodin) (p190)	0.69	−0.53	0.02
Q6PW52	GABA-A gamma2 long isoform (Gamma-aminobutyric acid A receptor, gamma 2) (Gamma-aminobutyric acid receptor subunit gamma-2)	0.68	−0.55	0.01
O35760	Isopentenyl-diphosphate Delta-isomerase 1 (EC 5.3.3.2) (Isopentenyl pyrophosphate isomerase 1) (IPP isomerase 1) (IPPI1)	0.68	−0.56	0.01
G3V679	Transferrin receptor protein 1 (Transferrin receptor, isoform CRA_a)	0.68	−0.56	0.02
P12839	Neurofilament medium polypeptide (NF-M) (160 kDa neurofilament protein) (Neurofilament 3) (Neurofilament triplet M protein)	0.66	−0.60	0.04
P62762	Visinin-like protein 1 (VILIP) (21 kDa CABP) (Neural visinin-like protein 1) (NVL-1) (NVP-1)	0.66	−0.60	0.00
G3V7U2	Microtubule-associated protein 1 A, isoform CRA_c (Microtubule-associated protein 1A)	0.65	−0.62	0.00
P63041	Complexin-1 (Complexin I) (CPX I) (Synaphin-2)	0.62	−0.69	0.00
F1LRZ7	Neurofilament heavy polypeptide	0.61	−0.71	0.01
P01830	Thy-1 membrane glycoprotein (Thy-1 antigen) (CD antigen CD90)	0.57	−0.81	0.00
G3V774	F-box only protein 2 (F-box protein 2)	0.56	−0.83	0.02
P19527	Neurofilament light polypeptide (NF-L) (68 kDa neurofilament protein) (Neurofilament triplet L protein)	0.54	−0.88	0.01
A0A0G2JZL9	Endothelial cell-selective adhesion molecule	0.53	−0.90	0.03
A0A0G2K0T6	Gamma-synuclein	0.38	−1.41	0.00
F1LUD3	AHNAK nucleoprotein 2	0.30	−1.74	0.01

**Table 3 ijms-22-08592-t003:** Protein regulation in the nasal quadrant 2 weeks after pONT compared to contralateral control (*p* < 0.05 and FC log2 > 0.43 (upregulation) or ≤ −0.43 (downregulation)). R = right; L = left; FC = fold change.

Uniprot	Protein Name	FC L/R	LogFC	*p*-Value
Upregulation				
Q5U2P2	Immunoglobulin superfamily member 11 (IgSF11)	3.17	1.67	0.02
D3ZZU8	Fructosamine 3 kinase (Similar to fructosamine-3-kinase (Predicted), isoform CRA_a)	2.47	1.31	0.02
Q6MGB4	H2-K region expressed gene 4, rat orthologue (RCG60794, isoform CRA_a) (Solute carrier family 39 (Zinc transporter), member 7) (Solute carrier family 39 member 7)	2.07	1.05	0.00
P47819	Glial fibrillary acidic protein (GFAP)	1.96	0.97	0.02
Q5BJQ2	Ubiquitin carboxyl-terminal hydrolase MINDY-1 (EC 3.4.19.12) (Deubiquitinating enzyme MINDY-1) (Protein FAM63A)	1.86	0.90	0.01
F1M118	“Obsolete”	1.85	0.89	0.02
D3ZKK3	Consortin, connexin sorting protein	1.70	0.77	0.04
D3ZPC4	Neural cell adhesion molecule L1	1.68	0.75	0.00
Q6QI79	Cap-specific mRNA (nucleoside-2′-O-)-methyltransferase 1 (LRRG00129)	1.60	0.68	0.03
F2Z3T8	Guanine nucleotide-binding protein subunit gamma	1.60	0.68	0.00
F1M9D6	Signal transducer and activator of transcription	1.58	0.66	0.05
P14056	Serine/threonine-protein kinase A-Raf (EC 2.7.11.1) (Proto-oncogene A-Raf) (Proto-oncogene A-Raf-1)	1.55	0.63	0.04
O88775	Embigin	1.54	0.62	0.02
Q6AYE2	Endophilin-B1 (SH3 domain-containing GRB2-like protein B1)	1.52	0.61	0.03
G3V836	Clusterin	1.52	0.60	0.03
P14480	Fibrinogen beta chain (Liver regeneration-related protein LRRG036/LRRG043/LRRG189) (cleaved into: Fibrinopeptide B; Fibrinogen beta chain)	1.51	0.60	0.01
Q9EPF2	Cell surface glycoprotein MUC18 (Gicerin) (Melanoma cell adhesion molecule) (Melanoma-associated antigen MUC18) (CD antigen CD146)	1.47	0.56	0.02
A0A0G2JUB0	Protein FAM3C	1.40	0.49	0.04
P62193	26S proteasome regulatory subunit 4 (P26s4) (26S proteasome AAA-ATPase subunit RPT2) (Proteasome 26S subunit ATPase 1)	1.39	0.48	0.01
M0R567	Interferon regulatory factor 2-binding protein-like	1.38	0.46	0.00
Downregulation				
P63041	Complexin-1 (Complexin I) (CPX I) (Synaphin-2)	0.66	−0.59	0.02
G3V7W1	Programmed cell death protein 6 (Apoptosis-linked gene 2 protein homolog) (ALG-2)	0.60	−0.73	0.01
Q62813	Limbic system-associated membrane protein	0.56	−0.82	0.00
D3ZDX5	Plexin B1	0.55	−0.87	0.04
D4ABV0	RGD1566320 (RGD1566320 (predicted))	0.43	−1.20	0.01

**Table 4 ijms-22-08592-t004:** Bioinformatic analysis using SWATH-based proteomic data obtained from temporal (T) and nasal (N) retinal quadrants at 2 weeks after pONT compared to the corresponding contralateral control (T/N). R = right eye; L = left eye.

	Differentially Expressed Genes	Genes with Measured Expression	Gene Ontology TERMS	miRNAs	Upstream Regulators	Chemical Upstream Regulators	Associated Diseases
LT vs. RT	59	2910	842	6	72	103	27
LN vs. RN	24	2901	339	0	30	72	12

**Table 5 ijms-22-08592-t005:** Pathway analysis of regulated proteins in temporal and nasal quadrants 2 weeks after pONT.

Pathway Name	Pathway ID	*p*-Value	*p*-Value (FDR)
LT vs. RT			
Ferroptosis	04216	5.332 × 10^−4^	0.063
HIF-1 signaling pathway	04066	0.002	0.133
Adipocytokine signaling pathway	04920	0.013	0.527
Necroptosis	04217	0.022	0.594
Cell adhesion molecules	04514	0.026	0.584
LN vs. RN			
Complement and coagulation cascades	04610	0.005	0.401
Hepatitis B	05161	0.018	0.408
Jak-STAT signaling pathway	04630	0.019	0.408
Pancreatic cancer	05212	0.024	0.408
Serotonergic synapse	04726	0.030	0.408

**Table 6 ijms-22-08592-t006:** Gene ontology analysis of temporal and nasal quadrants 2 weeks after pONT.

Results	LT vs. RT			LN vs. RN		
	GO Term	*p*-Value	*p*-Value (FDR)	GO Term	*p*-Value	*p*-Value (FDR)
Molecular functions	Signaling receptor binding	2.600 × 10^−4^	0.054	mRNA (nucleoside-2′-O-)-methyltransferase activity	0.008	0.222
	Protein-containing complex binding	2.800 × 10^−4^	0.054	Fructosamine-3-kinase activity	0.008	0.222
	Thyroid hormone binding	0.001	0.130	Cell adhesion molecule binding	0.012	0.222
	Structural constituent of postsynaptic intermediate filament cytoskeleton	0.001	0.130	Signaling receptor binding	0.013	0.222
	Tau protein binding	0.002	0.130	PDZ domain binding	0.014	0.222
Biological processes	Intermediate filament organization	2.000 × 10^−7^	5.018 × 10^−4^	Homophilic cell adhesion via plasma membrane adhesion molecules	2.700 × 10^−4^	0.207
	Supramolecular fiber organization	6.300 × 10^−7^	5.854 × 10^−4^	Positive regulation of multicellular organismal process	6.900 × 10^−4^	0.207
	Neuron projection regeneration	7.000 × 10^−7^	5.854 × 10^−4^	Cell adhesion	0.001	0.207
	Intermediate filament-based process	6.600 × 10^−6^	0.002	Biological adhesion	0.001	0.207
	Intermediate filament cytoskeleton organization	6.600 × 10^−6^	0.002	Cell-cell adhesion via plasma-membrane adhesion molecules	0.001	0.207
Cellular components	Postsynaptic intermediate filament cytoskeleton	3.400 × 10^−5^	0.013	Side of Membrane	0.002	0.310
	Neurofibrillary tangle	8.400 × 10^−5^	0.015	External side of plasma membrane	0.007	0.310
	Axon	1.700 × 10^−4^	0.018	Spherical high-density lipoprotein particle	0.008	0.310
	Schaffer collateral—CA1 synapse	3.900 × 10^−4^	0.018	Cytoplasmic side of lysosomal membrane	0.008	0.310
	Neuronal cell body	4.100 × 10^−4^	0.018	Perinuclear endoplasmic reticulum lumen	0.008	0.310

## Data Availability

The data presented in this study are available in this article and Appendix A section.

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
