# Peer review of "Differential Retinal Protein Expression in Primary and Secondary Retinal Ganglion Cell Degeneration Identified by Integrated SWATH and Target-Based Proteomics"

_ijms, 2021, doi:10.3390/ijms22168592_

Round 1

Reviewer 1 Report

The authors aim to identify the retinal proteins associated with primary and secondary retinal ganglion cell (RGC) degeneration and explore their molecular pathways. The authors have used SWATH label-free and tageted-based mass spectrometry (MRM) to identify the proteomes in various retinal locations in response to localized optic nerve injury.

The study is of high relevance in the field, however the bioinformatics approach needs to be improved.

Line 103-104 Was this difference statistically evaluated? Would be nice to see the evidence as supplemental material. Evidence of no difference are equally important in association studies.

Line 150-152. String reports different type of evidences for protein-protein interactions (PPI). Howerver, there are other sources of PPI such as http://www.interactome-atlas.org/download and BioPlex 2021 https://www.biorxiv.org/content/10.1101/2020.01.19.905109v1 

that contain most recent interactions protein partners.If conclusions needs to be drawn based on PPI the authors need to re-consider the network-based approach accounting form the resources above and include them together with String. After the background information, here the PPI, have been properly derived what ever observations of interactions need to be tested against their random probability (as shown for example by Barabasi et al, Science 347:6224, 1257601-1 (2015)) to exclude that conclusion are just random.

Line 506  Specify what kind of t-test and what are the assumptions made that explain why this is the best modelling strategy for group differences.

Line 510 (paragraph 4.7) This paragraph needs to describe what kind of statistical methods and statistical models have been used to assess the differential expressed proteins. The expression: "The data was analysed in the context of pathways" does not mean anything. Data, protein expression or list of genes can be used to statistically determine the enrichment of specific pathways that can linked to the MoA or phenomena we are trying to describe. Please use a more rigorous description on the pathway analysis.

The table with all proteins quantification needs to be in excel format to make sure analysis are reproducible. 

The subset of tables that are used to drawn conclusions can be in pdf within the SI but I strongly recommend to also provide the excel format for these results too.

I suggest to consider to submit their data following the criteria  

https://www.ebi.ac.uk/pride/archive/ or here http://www.proteomexchange.org 

Reviewer 2 Report

The manuscript pdf likely have not assembled correctly. This reviewer's downloaded version lacks section 5. conclusions and figures. Without them a proper review cannot be performed. Also recommend submitting the dataset in PRIDE or an appropriate database. 

The manuscript is timely and important to the readership but again without all parts of the manuscript the review cannot be performed. 

Round 2

Reviewer 1 Report

The authors have magnificently tackled all the raised points. Thank you for your dedication and scientific rigour.

Author Response

We greatly appreciate Reviewer 1 for the time and effort on reviewing our manuscript. 

Reviewer 2 Report

The manuscript despite being timely and interesting is not yet ready for a serious review. Major modifications are necessary before the reviewers attention is drawn for a careful review. Table 1 states some Marker view, in the entire manuscript there is no description of the same. Why is table 1 necessary is unclear. This table do not add anything for the readership and appears to be an internal accounting not a summary for the readership. In the Table 2, the entry begins with an obsolete accession: M0R9G2. Why include this obsolete entry at the first place? In summary, a carefully reviewed and edited manuscript with improved presentation should be submitted to avoid wasting a serious reviewers time. The manuscript in its current form do not warrant an experienced reviewer's attention. 

Round 3

Reviewer 2 Report

The authors continue to peddle a manuscript merely with inappropriate "cosmetic" changes. M0R9G2, the accession number in the uniprot database shows an obsolete protein. It is not Alpha-2-macroglobulin. The problems identified in previous round of review continues to plague the manuscript. Table 2 needs a careful revision, not cosmetic change. Authors have altered the numbers of proteins. They should do a careful review and submit a fresh manuscript.
